# Strong plates enhance mantle mixing in early Earth

Roberto Agrusta[1,3], Jeroen van Hunen [1] & Saskia Goes [2]

In the present-day Earth, some subducting plates (slabs) are flattening above the upper–lower mantle boundary at ~670 km depth, whereas others go through, indicating a mode between layered and whole-mantle convection. Previous models predicted that in a few hundred degree hotter early Earth, convection was likely more layered due to dominant slab stagnation. In self-consistent numerical models where slabs have a plate-like rheology, strong slabs and mobile plate boundaries favour stagnation for old and penetration for young slabs, as observed today. Here we show that such models predict slabs would have penetrated into the lower mantle more easily in a hotter Earth, when a weaker asthenosphere and decreased plate density and strength resulted in subduction almost without trench retreat. Thus, heat and material transport in the Earth's mantle was more (rather than less) efficient in the past, which better matches the thermal evolution of the Earth.

---

[1] Department of Earth Sciences, Durham University, DH1 3LE Durham, UK. [2] Department of Earth Science and Engineering, Imperial College London, SW7 2AZ London, UK. [3] Present address: Laboratoire de Géologie de Lyon, Université de Lyon, École Normale Supérieure de Lyon, 69007 Lyon, France. Correspondence and requests for materials should be addressed to R.A. (email: roberto.agrusta@ens-lyon.fr)

Seismic imaging of Earth's mantle has shown that when subducting plates reach the upper–lower mantle boundary at ~670 km depth, they can either penetrate straight into the lower mantle or flatten in the mantle transition zone above this boundary[1,2] (Fig. 1a). How easily slabs penetrate into the lower mantle exerts a key control on the efficiency of mass and heat flux across the mantle between the surface and the boundary with the outer core, to which active upwellings probably contribute only 10–20% to the total heat transport[3,4]. The mix of temporarily stagnant and penetrating slabs in the mantle transition zone indicates that the present-day mantle is in a transitional mode between layered and whole convection[5,6]. However, convection style might have changed during the Earth's history as mantle temperatures decreased by 200 °C–300 °C from the Archean eon to the present[7,8], and previous convection studies predict that this mantle cooling would switch convection style from a dominantly layered system in the past to a system intermediate between whole and layered at the present day[6,9,10] (Fig. 1b).

The upper–lower mantle boundary coincides with the endothermic phase transition in the main mantle mineral olivine (ol), from its ringwoodite (rg) phase to its denser post-spinel assemblage (perovskite and magnesiowustite, pv + mw), and it likely also localises at least part of the factor 10–100 viscosity increase from upper to lower mantle[11,12]. This phase transition gets depressed to larger depths inside the cold slab from its equilibrium depth (~670 km) and might hamper the flow across it. This deflection depends on the phase transition Clapeyron or pressure–temperature slope and if the Clapeyron slope is strong (negative) enough, it can break mantle convection into two layers[6,9,13,14]. Whether convection is layered or not depends on whether the positive-phase buoyancy of the endothermic transition exceeds the negative thermal buoyancy of the slabs, and it has been demonstrated that the necessary critical buoyancy number $P$ (the ratio of the phase and thermal buoyancy, eq. 5) to induce layered convection by the endothermic phase transition decreases with increasing convective vigour, i.e., increasing mantle temperature (Rayleigh number, eq. 4)[5,6,9] (Fig. 1b). This

stronger propensity for layering at higher Rayleigh number has been attributed to the lower viscosity and smaller scale of down- and upwellings in a hotter mantle[9,15], which makes the transmission of the thermal buoyancy forces, necessary to overcome the effect of an endothermic phase transition, less efficient. This was found to hold in both models with an isoviscous mantle[6,9,13] and in models that test the effect of temperature-dependent and/or stress-dependent viscosity, which leads lithosphere and slabs to behave more plate-like[5,15]. Thus, it is generally assumed that the previously hotter mantle convected in a more layered style.

On Earth, its observed that older (denser and stronger) plates have a higher tendency to produce trench retreat and flat slabs above the upper–lower mantle boundary around ~670 km depth than young plates[16,17]. This behaviour is reproduced in recent dynamical models where plate boundaries move in response to the slab dynamics. In these models[2,18–21], stronger and denser (old) slabs interacting with both an endothermic phase change and viscosity increase induce trench retreat and stagnate (at least for 10s to 100s of m.y.), whereas weaker and lighter (young) slabs accumulate at relatively stationary trenches, which aids penetration. Although other factors, e.g., the persistence of metastable phases in the slab's coldest core and associated slab weakening[16,22–24], may additionally hamper the sinking of older slabs through the transition zone, variable plate age at the trench can explain the primary observations of today's mixed slab-transition-zone dynamics and its relation to trench motion[2,20]. In this study, we use these calibrated models to re-examine how such more plate-like and mobile slabs behave under hotter mantle conditions. The new results show that, contrary to previous work, higher mantle temperatures favour less layered convection with decreased slab stagnation in the transition zone, which has important consequences for Earth's evolution.

## Results

**Mantle and plate parameters investigated.** In this study, it has been performed a set of 35 numerical simulations with the dynamically self-consistent thermo-mechanical two-dimensional

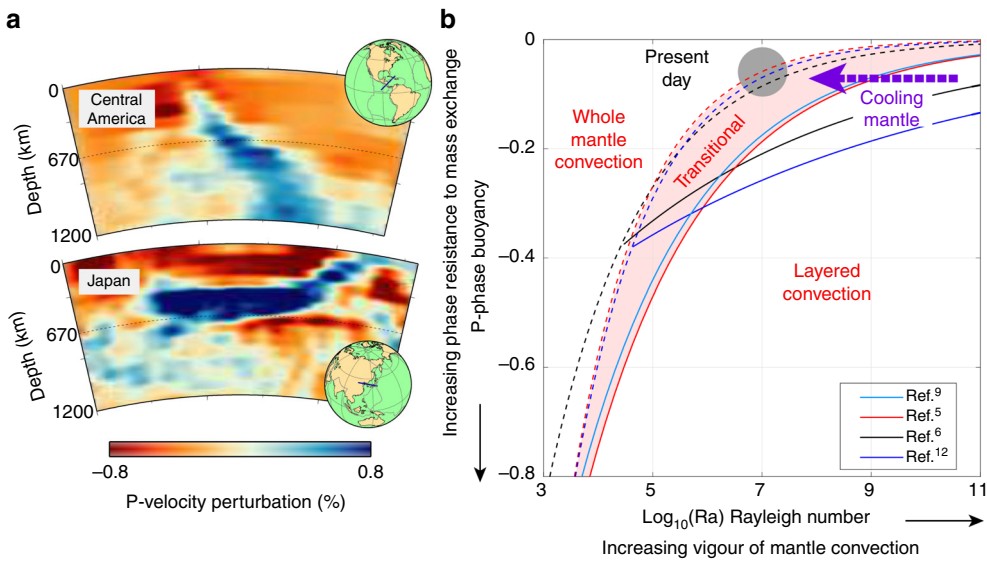

**Fig. 1** Present and previously predicted mantle convection styles. **a** Examples from seismic tomography of a slab readily penetrating the transition zone (below Central America) and a slab that has ponded (below Japan), obtained using the 3D MIT-P08 seismic velocity model of Ref. [69]. **b** Regime diagram showing how previous studies[5,6,9,13] predict that the style of mantle convection varies with buoyancy number $P$ (the ratio of the phase buoyancy of the endothermic phase transition hampering slab sinking over the thermal buoyancy which drives slab sinking) and Rayleigh number (the ratio of convection-driving over convection-resisting forces, which increases proportionally to mantle temperature). All studies agreed that the critical phase buoyancy required to layer convection decreases (becomes less negative) with increasing Rayleigh number. The grey circle represents the estimated present-day conditions [from Ref. [6]] and the arrow the likely change from layered convection in an early Earth to a transitional mode today

(2D) subduction models of Agrusta et al.[18]. (see Methods, Supplementary Table 1) to investigate how old (100 Myr) and young (50 Myr) plates interact with a phase and viscosity boundary at different mantle temperatures. Mantle potential temperatures (i.e., temperatures at the top of the convective mantle geotherm, the mantle adiabat) are varied from 50 °C cooler to 200 °C hotter than the present day. This results in a mantle viscosity jump at the upper–lower mantle boundary between a factor of 10 (at present-day conditions) to a factor of 40 (in the hotter mantle). The models include the two main ol phase transitions, the exothermic ol-wadsleyite (ol-wd) transition at ~410 km depth, and the rg-pv + mw at ~670 km. To test the effect of phase buoyancy $P$, the Clapeyron slopes have been varied over a plausible range, from 3 MPa·K$^{-1}$ to 5 MPa·K$^{-1}$ (ol-wd)[25,26] and from $-1$ MPa·K$^{-1}$ ($P = -0.036$) to $-3$ MPa·K$^{-1}$ ($P = -0.109$) (rw-pv + mw)[27,28]. The models presented use a Newtonian rheology and assume a composition of 100 wt% of ol, but additional models, with a composite non-Newtonian creep and only 60 wt% of ol, which display the same styles of behaviour, are in the Supplementary Figure 2,3. In three additional simulations, the effect of slab strength at transition zone depths has been investigated, by reducing slab viscosity below 400 km depth.

**Present-day subduction dynamics.** The present-day models produce the mixed stagnation-penetration style where older, colder plates have a stronger tendency to stagnate and younger plates to penetrate[16,17]. Figure 2a,b illustrates how a young, hot, and weak subducting plate drives only modest trench retreat and therefore penetrates directly into the lower mantle, whereas an old, cold and strong plate sinks with significant trench retreat and flattens in the transition zone[2,18].

A useful measure for slab penetration into the lower mantle is to compare the accumulated volume of slab material in the transition zone ('Slab$_{TZ}$') and the lower mantle ('Slab$_{LM}$') through time (Fig. 2c). In the stagnant case, Slab$_{TZ}$ increases more quickly than Slab$_{LM}$, because a significant part of the slab accumulates in the transition zone. In contrast, for the penetrating young slab, the amount of slab material that collects in the transition zone is low and almost constant during the simulation time. This behaviour can be summarised by a slab accumulation rate ($D$) in each mantle layer ($D_{TZ}$ and $D_{LM}$), calculated as:

$$D_{TZ/LM} = \frac{\text{Slab}_{TZ/LM}(t_{end}) - \text{Slab}_{TZ/LM}(t_{670})}{\text{Time}(t_{end}) - \text{Time}(t_{670})} \quad (1)$$

where $t_{670}$ and $t_{end}$ correspond to the model time at which the slab reaches the base of the upper mantle (670 km) and the end time of the simulation, respectively. The ratio $D_{TZ}/D_{LM}$ is used to classify slab penetration or stagnation, with values > 1 for significantly stagnating slabs and < 1 for mostly penetrating slabs.

**Dynamics in a hotter mantle.** The effects of a hotter mantle are shown in Fig. 3 through snapshots of old-slab simulations at two different model times for the most negative Clapeyron slope, i.e., the cases most likely to stagnate (for times of 80 m.y. or longer). At present-day temperatures ($\Delta T_{pot} = 0$ °C, Fig. 3a), the slab flattens at the base of the upper mantle similar to the case shown in Fig. 2b, as can be seen in the evolution of Slab$_{TZ}$ and Slab$_{LM}$ (Fig. 3e, f, blue lines), with lower-mantle slab penetration even more reduced due to the stronger Clapeyron slope. At higher mantle temperatures (Fig. 3b–d), the slab folds and piles up in the transition zone. When a sufficiently large volume of slab has accumulated in the transition zone, its negative buoyancy is able to overcome the phase resistance, and the slab starts sinking into the lower mantle. Figure 3e, f further illustrate how the slab initially accumulates in the transition zone (Slab$_{TZ}$ increasing),

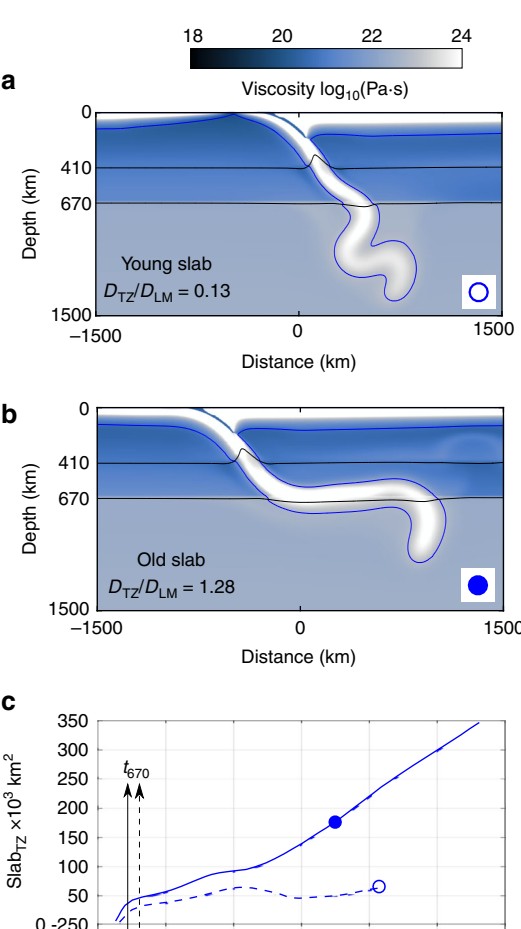

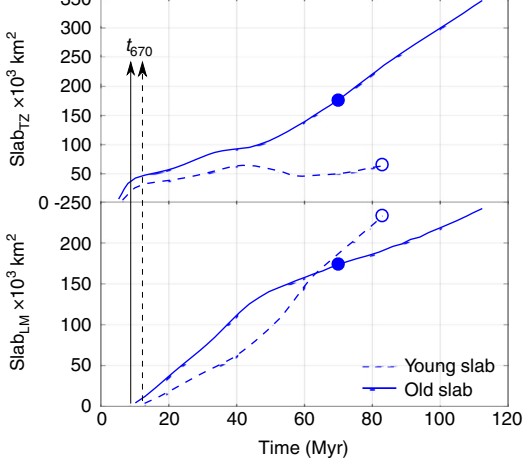

**Fig. 2** Slab dynamics for present-day mantle temperature. Cases shown are a young penetrating slab (Simulation 11, open blue circle) (**a**) and old flattened slab (Simulation 12, solid blue circle, see Supplementary Table 1) (**b**). The initial trench is located at $x = 0$ km. The blue line is the contour delimiting the slab at constant potential temperature of 1300 °C. For both cases the value of the slab accumulation rate $D_{TZ}/D_{LM}$ is indicated. **c** Evolution of the volume of slab material in transition zone (Slab$_{TZ}$) and lower mantle (Slab$_{LM}$). The dots mark the time of the snapshots in **a**, **b**. Although significant volumes of the old slab (solid lines) accumulate in the transition zone, most of the young slab material (dashed lines) goes straight through. The arrows mark the times at which the slabs reach the 670 km depth ($t_{670}$)

followed by a relatively stable phase where slab material slowly increases in the lower mantle (Slab$_{LM}$ increasing), and a final stage in which Slab$_{TZ}$ decreases and the slab sinks more rapidly into the lower mantle. The time towards this accelerated slab lower-mantle sinking decreases with increasing Rayleigh number.

The slab accumulation rates $D_{TZ}$ and $D_{LM}$, and their ratios for all cases, including different slab ages and Clapeyron slopes are compiled in Supplementary Figure 1. All slabs in the hotter mantle models ($\Delta T_{pot} = 100$ °C and 200 °C), irrespective of the initial slab ages, have high $D_{LM}$ and low $D_{TZ}$, and $D_{TZ}/D_{LM} < 1$,

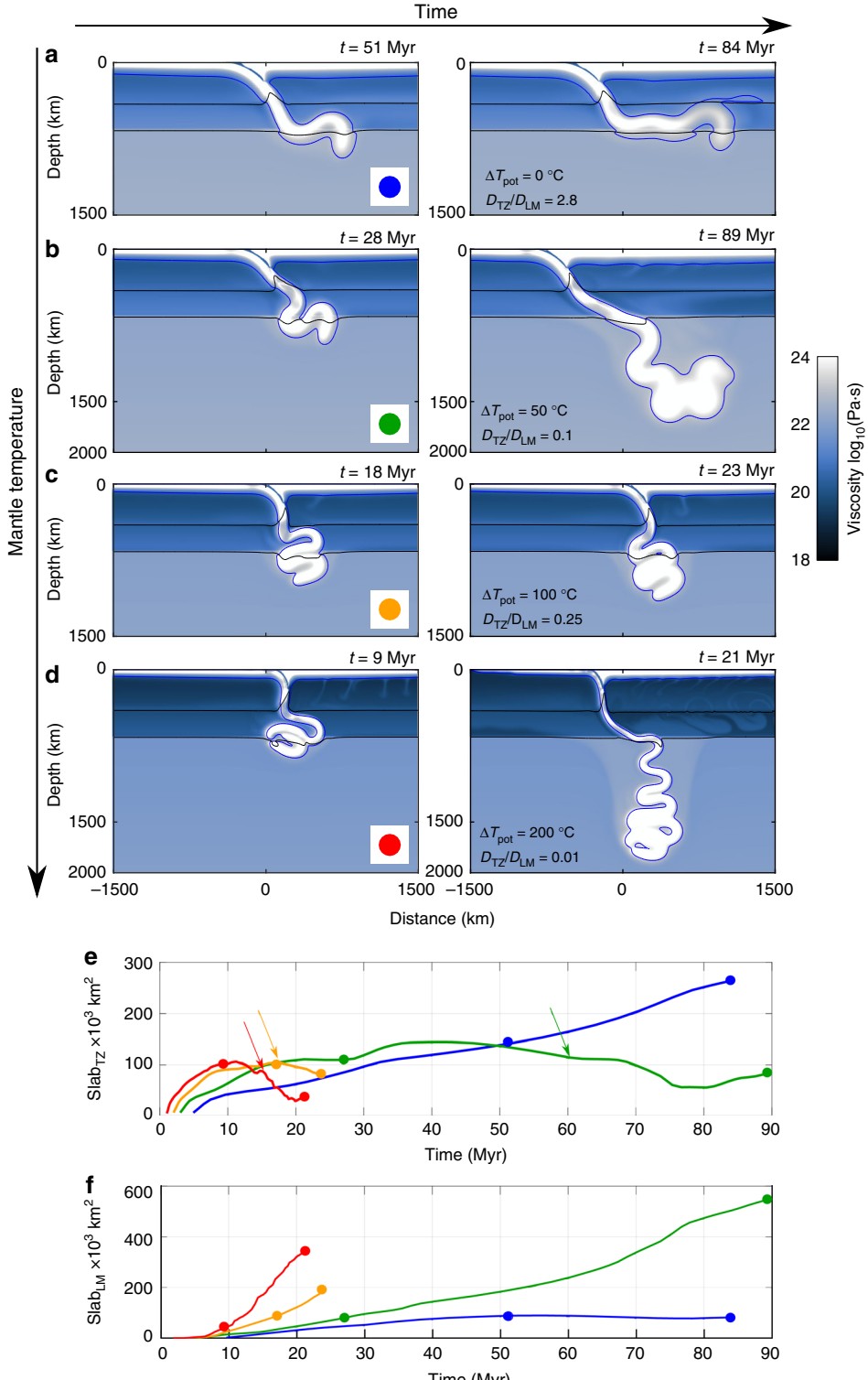

**Fig. 3** Penetrating slabs in a hotter mantle. **a–d** Old (initial age 100 Myr) slab evolution for different mantle temperatures (illustrated by two snapshots each) at strongly negative post-spinel Clapeyron slope ($-3$ MPa•K$^{-1}$) (Simulations 16, 20, 26 and 32, Supplementary Table 1): **a** present-day temperature, $\Delta T_{pot} = 0$ °C (blue), **b** $\Delta T_{pot} = +50$ °C (green), **c** $\Delta T_{pot} = +100$ °C (orange), **d** $\Delta T_{pot} = +200$ °C (red). This colour coding is subsequently used in **e**, **f**, and Fig. 4. For each case the value of $D_{TZ}/D_{LM}$ is indicated. The evolution of slab material in the transition zone Slab$_{TZ}$ (**e**) and lower mantle Slab$_{LM}$ (**f**). The arrows in **e** indicate the approximate timing of accelerated lower-mantle slab sinking events in the hotter mantle models. The dots mark the timing of the snapshots in **a–d**

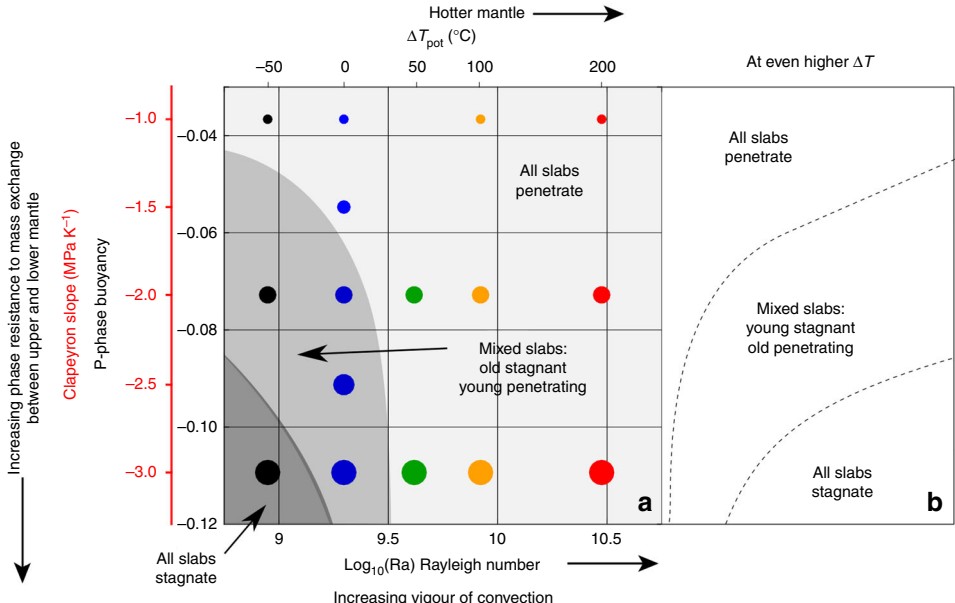

**Fig. 4** Mantle mixing modes at different mantle temperatures. Regime diagram obtained from all our simulations (coloured dots) as a function of the phase buoyancy number $P$ (and corresponding Clapeyron slope) for the endothermic phase transition and Rayleigh number (and corresponding mantle temperature). Note that the Ra of our regional models are not directly comparable to those of the global-scale models in Fig. 1b, but present-day Earth conditions are likely somewhere around the middle blue dot. Dark grey field covers the domain of pure slab stagnation, middle grey the domain for young slab penetration and old slab stagnation, light grey field the domain of pure slab penetration. On the right side, the regime diagram is schematically extended as expected at even higher temperature from the results of our weak slabs models together with results from previous models where plates were weak and trenches less mobile

indicating easy penetration into the lower mantle. In contrast, slabs in a colder mantle ($\Delta T_{pot} = 0\,°C$ and $-50\,°C$) have lower $D_{LM}$ and higher $D_{TZ}$, and some stagnate in the transition zone, while others penetrate easily. For weaker phase-transition resistance (less negative Clapeyron slope values), all slabs tend to penetrate directly into the lower mantle, whereas for more negative Clapeyron slope values, easy stagnation occurs for plates with old initial ages.

Figure 4 summarises these results in a regime diagram of slab-transition zone interaction style as a function of Rayleigh and phase buoyancy numbers, similar to what was done in previous studies[5,6,13] (Fig. 1b). Layered convection, where slab stagnation occurs for both young and old plates, is only achieved at low Rayleigh number (i.e., cooler Earth than today) and low phase buoyancy number (most negative Clapeyron slope). At intermediate $P$ and Ra, both modes are found, with easy penetrating young and long temporal stagnant old slabs. At higher Ra (hotter Earth), no slab stagnation is observed. Note that these boundaries can shift within the uncertainties and trade-offs between model parameters. At a higher viscosity jump at the base of the transition zone, the field of stagnant and mixed modes expands to lower phase buoyancy and higher Ra. A reduction of the asthenospheric mantle viscosity, leading to less trench mobility[29,30], would induce an opposite shift. The main features of the regime diagram as a function of temperature are however robust.

In a cooler Earth, older stronger slabs are able to drive trench retreat, which lays out the slabs in the transition zone, hampering their entrance into the lower mantle. At higher temperatures, trench retreat is discouraged by lower slab strength, which facilitates plate bending, and decreases asthenospheric viscosities, which inhibits trench retreat[31,32]. These factors, together with a lower resistance from a hotter, and therefore less viscous lower mantle, allow slabs in a hotter mantle to enter the lower mantle more easily than at modern mantle temperatures. Given that

today's mantle is in a mixed mode, these models imply that mantle cooling increases the occurrence of slab stagnation, and, in contrast to what was found in earlier studies, in a hotter Earth, slab penetration would have been dominant.

**Slab strength**. Slab weakening in the transition zone, which can be due to grain-size reduction during phase transformation, has been previously proposed to lead to slab stagnation[16]. The slabs presented here are stronger than in previous models, allowing them to penetrate the endothermic phase transition even when the mantle temperature is increased by 200 degrees. To investigate whether slab strength accounts for this different model behaviour, one of the models has been re-run with a weaker slab in and below the transition zone, by reducing the maximum viscosity below 400 km depth to $10^{23}$ Pa s, $5 \times 10^{22}$ Pa s and $10^{22}$ Pa s (Fig. 5). These weaker slabs deform considerably when they reach the bottom of the upper mantle, spreading out in the transition zone. The weakest slabs, with viscosities of only a few times the background mantle viscosity, fail to enter the lower mantle (Fig. 5c). Most likely, previous studies that concluded that stagnation increases with Ra, investigated a regime where already weak slabs become even weaker under hotter mantle conditions, which leads to increasing stagnation. This behaviour might be expected for the presented models as well if mantle temperature is raised further (i.e., for much higher Ra) (Fig. 4). In an intermediate regime between hot penetrating slabs and even hotter stagnant ones, the weakest (youngest) slabs would stall while the colder stronger ones would still be able to penetrate, opposite to what happens in the mixed mode of present-day models[18] and what is observed on Earth[2,16,17].

## Discussion

Several factors not accounted for in this work may affect slab dynamics in a hotter mantle, most notably the effects of a higher

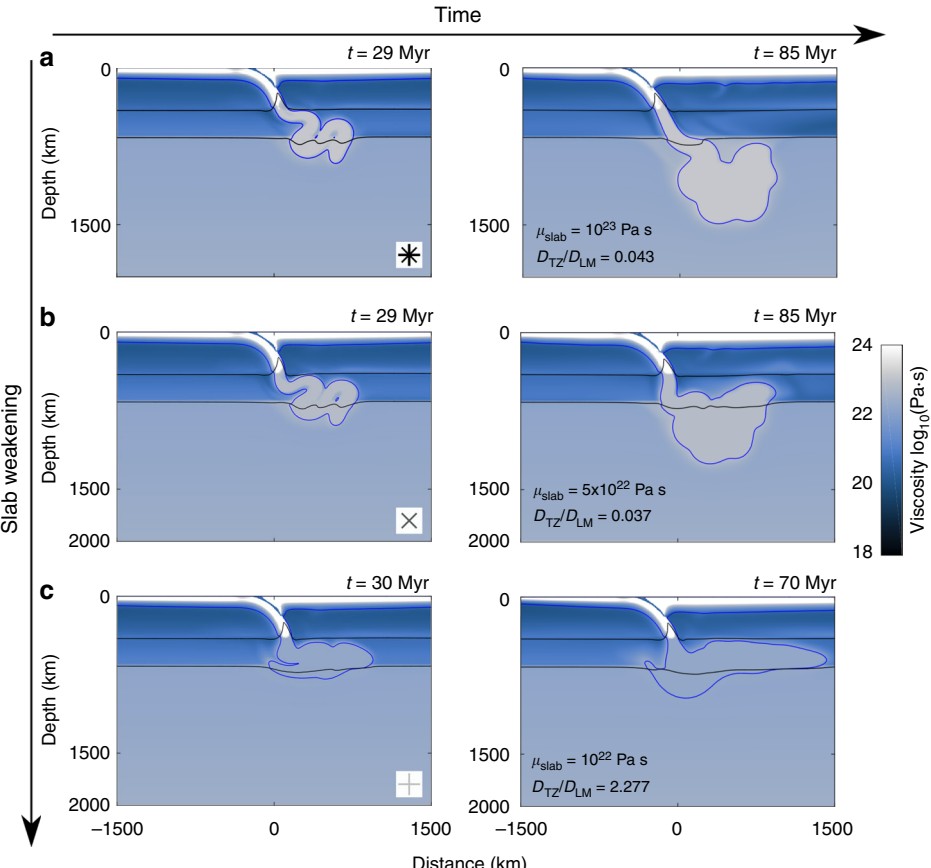

**Fig. 5** Weaker slab interaction with the upper–lower mantle boundary. Slab evolution of simulations 20a,b,c in which the maximum viscosity cut-off below 400 km depth is reduced to: **a** $10^{23}$, **b** $5 \times 10^{22}$ and **c** $10^{22}$ Pa s. For the three cases the value of $D_{TZ}/D_{LM}$ is indicated, highlighting how very weak transition-zone slabs tend to stagnate

melting degree on plate buoyancy and strength[33,34]. High mantle temperatures in the past could have produced thicker oceanic crust at mid-ocean ridges[35] and leave behind a water-depleted stiffer lithosphere. Based on previous studies, possible implications are discussed.

A thicker and more buoyant crust would probably resist subduction, similar to modern aseismic ridges, but may not prevent it[36], and could have made subduction episodic[33]. Moreover, a lower-density, but still subductable crust leads to a subduction style that would look like continental subduction, in which trench retreat is usually absent[37], and hence encourages penetration[18,19]. Some studies suggest that the early Earth oceanic crust compositions would have been denser than ambient mantle[38], which instead would have further have facilitated subduction and probably lower-mantle penetration.

A lithosphere that is substantially dehydrated upon melt extraction may be between a factor 2–3 to 100 stronger than hydrated plates[39–41]. The effect of strengthening by dehydration may be partially or totally negated by melt weakening[42], or rehydration in bending faults at the trench[43]. Note that when slabs become significantly stronger than the present-day effective slab bending strength, subduction will stop completely, because plate bending can no longer be achieved with the available slab potential energy[44]. Hence, during the time over which subduction has been active, plates were probably never more than a few times stronger than those at the present day. However, even if slab strength decreased less rapidly with increasing mantle temperature than in the presented models because of a trade-off with strengthening by dehydration, a weaker lower mantle would still enhance penetration[18,45] and the transitional mode may prevail

to somewhat higher temperatures than the presented models predict before all slabs start to penetrate.

Metastable phases inside the coldest slabs have been proposed to contribute to the stagnation of older plates in the transition zone[16,17,22,24]. However, at higher temperatures, both the effects of metastability and concurrent slab weakening due to grain-size reduction will be suppressed, thus also facilitating slab penetration in a hotter Earth.

Presently, subducting slabs exhibit mixed behaviour in the transition zone, where older plates have a tendency to stagnate, while younger ones penetrate easily into the lower mantle. Models in which slabs have plate-like rheology and trenches are mobile reproduce this behaviour and show that slabs would have been sinking more easily into the lower mantle in a hotter, earlier Earth. This would have allowed the early Earth to cool and mix mantle heterogeneities more efficiently than occurs at the present-day. Some studies have argued that dense piles in the deep mantle, suggested to be the cause of the seismic large-low shear velocity provinces, have been in stable locations for half a billion years or more[46,47]. This is difficult in a system of efficient whole-mantle convection[48–51], such as the presented models predict for much of Earth evolution.

The presented study does ignore the active upwelling part of the global convection. As mentioned in the introduction, upwellings probably contribute only ~10–20 % of present-day mantle heat flux[3,4]. Furthermore, upwellings are expected to readily cross the phase transition at hotter mantle conditions, because for transition-zone temperatures higher than 2000 °C, the transition in a pyrolite or harzburgite composition to post-spinel phases at the base of the upper mantle becomes exothermic

(positive Clapeyron slope)[52], which facilitates material flow through the phase transition.

The presented results contrast with previous modelling studies[5,6,9,13,15] that predicted that, in a hotter mantle, the phase and viscosity changes at the base of the mantle transition zone would have increasingly hampered slab sinking into the lower mantle and thus would have led to layered convection in upper and lower mantle. The behaviour that the presented models predict reconciles dynamics with cooling history calculations. Layered convection would not have cooled the early Earth efficiently enough to explain present day heat flow and mantle temperature[53]. Davies[54] proposed full mantle layering in an early Earth that would periodically collapse into catastrophic mantle overturns, a mechanism that would have allowed cooling in spite of layering. Instead, with the new results slab sinking into the lower mantle may have happened efficiently by regional lower-mantle sinking events.

Thus, early-Earth slabs probably favoured lower-mantle penetration and promoted whole-mantle convection. However, before plate tectonics started, perhaps around 3 Ga[55–57], down-wellings were probably more random, in the form of small-scale features[58] and this would have made mass exchange between upper and lower mantle less efficient. Consequently, the Earth may have undergone more mixing throughout its 'middle ages' and less so in its 'youth' and 'old age'.

## Methods

**Governing physics.** The slab-transition zone interaction is studied with 2D self-consistent subduction simulations using the finite-element code CITCOM[59–61]. The code solves the system of conservation of mass, momentum, and energy equations, for an incompressible fluid, at infinite Prandtl number, under the extended Boussinesq approximation[9], without internal heating.

The mantle phase transitions are included using a harmonic phase function[22]. The relative fraction of the heavier phase is described by the phase function $\Gamma$, varying from 0 and 1 as a function of pressure and temperature, as:

$$\Gamma_i = 0.5\left[1 + \sin\left(\frac{z - z_i - \gamma_i(T - T_i)}{d_i}\right)\right], \qquad (2)$$

where $d_i$ is the width of the transformation in depth, $\gamma_i$ is the Clapeyron slope, and $z_i$ and $T_i$ are the depth and temperature of the $i$th mantle phase transition at equilibrium conditions, respectively. $z$ and $T$ are depth and temperature.

The rheological model is assumed to be a combination of linear diffusion creep ($\mu_{diff}$) and a pseudo-brittle yield stress rheology ($\mu_y$). The effective viscosity $\mu_{eff}$ is calculated from the viscosities of the individual mechanisms as:

$$\mu_{eff} = \min(\mu_{diff}, \mu_y), \qquad (2.a)$$

with

$$\mu_{diff} = \Delta\mu_{lower/upper} A_{diff} \exp\left(\frac{E_{diff} + PV_{diff}}{RT}\right) \qquad (2.b)$$

and

$$\mu_y = \frac{\min(\sigma_0 + f_c P, \sigma_{max})}{\dot{\varepsilon}_{II}}. \qquad (2.c)$$

The factor $\Delta\mu_{lower/upper}$ defines the viscosity jump at 670 km depth and reduces to 1

## Table 1 List of parameters

| Symbol | Meaning | Unit | Value |
|---|---|---|---|
| **Global parameters** | | | |
| $H$ | Box height | km | 3000 |
| $\Delta T$ | Potential temperature drop | K | 1300 |
| $T_{pot}$ | Potential temperature | K | $1573 + \Delta T_{pot}$ |
| $\Delta T_{pot}$ | Temperature increase | K | ($-50$ to 200) |
| $\rho_0$ | Surface reference density | kg m$^{-3}$ | 3300 |
| $g$ | Gravity | m s$^{-2}$ | 9.8 |
| $\alpha_0$ | Surface thermal expansion | K$^{-1}$ | $3 \times 10^{-5}$ |
| $\kappa$ | Thermal diffusivity | m$^2$ s$^{-1}$ | $10^{-6}$ |
| $\mu_0$ | Reference viscosity | Pa s | $\mu_{eff}(z = 0, T = T_{pot})$ |
| $C_P$ | Heat capacity | J kg$^{-1}$ K$^{-1}$ | 1250 |
| $R$ | Gas constant | J mol$^{-1}$ K$^{-1}$ | 8.314 |
| **Rheological model parameters** | | | |
| Diffusion creep | | | |
| $A_{diff}$ | Pre-exponential upper mantle | Pa s | $1.87 \times 10^9$ |
| | Pre-exponential lower mantle | | $2.29 \times 10^{14}$ |
| $E_{diff}$ | Activation energy upper mantle | J mol$^{-1}$ | $3 \times 10^5$ |
| | Activation energy lower mantle | | $2 \times 10^5$ |
| $V_{diff}$ | Activation volume upper mantle | m$^3$ mol$^{-1}$ | $5 \times 10^{-6}$ |
| | Activation volume lower mantle | | $1.5 \times 10^{-6}$ |
| $\Delta\mu_{lower/upper}$ | Viscosity jump | - | 10 |
| Byerlee's plastic deformation | | | |
| $f_c$ | Friction coefficient | - | 0.2 |
| $\sigma_{max}$ | Maximum yield strength | MPa | 300 |
| $\sigma_0$ | Surface yield strength | MPa | 20 |
| **Mantle phase transition parameters** | | | |
| $\gamma_{ol-wd}$ | Clapeyron slope ol-wd transition | MPa K$^{-1}$ | (2.5 to 5) |
| $\gamma_{rw-pv+mw}$ | Clapeyron slope rg-pv+mw transition | MPa K$^{-1}$ | ($-0.5$ to $-3$) |
| $z_{ol-wd}$ | Central ol-wd transition depth | km | 410 |
| $z_{rw-pv+mw}$ | Central rw-pv+mw transition depth | km | 670 |
| $d_{ol-wd}$ | ol-wd transition width | km | 20 |
| $d_{rw-pv+mw}$ | rg-pr+mw transition width | km | 20 |
| $T_{ol-wd}$ | ol-wd transition potential temperature | K | $T_{pot}$ |
| $T_{rw-pv+mw}$ | rg-pr+mw transition potential temperature | K | $T_{pot}$ |
| $\Delta\rho_{ol-wd}$ | ol-wd transition density contrast | kg m$^{-3}$ | 250 |
| $\Delta\rho_{rw-pv+mw}$ | rg-pr+mw transition density contrast | kg m$^{-3}$ | 350 |

**Table 2 List of rheological parameters**

| Symbol | Meaning | Unit | Value |
|---|---|---|---|
| Diffusion creep | | | |
| $A_{diff}$ | Pre-exponential upper mantle | Pa s | $6.47 \times 10^9$ |
| | Pre-exponential lower mantle | | $1.87 \times 10^{14}$ |
| $E_{diff}$ | Activation energy upper mantle | J mol$^{-1}$ | $3 \times 10^5$ |
| | Activation energy lower mantle | | $2 \times 10^5$ |
| $V_{diff}$ | Activation volume upper mantle | m$^3$ mol$^{-1}$ | $4 \times 10^{-6}$ |
| | Activation volume lower mantle | | $1.5 \times 10^{-6}$ |
| Dislocation creep | | | |
| $A_{disl}$ | Pre-exponential upper mantle | Pa$^n$ s | $5 \times 10^{16}$ |
| $E_{disl}$ | Activation energy upper mantle | J mol$^{-1}$ | $5 \times 10^5$ |
| $V_{disl}$ | Activation volume upper mantle | m$^3$ mol$^{-1}$ | $11 \times 10^{-6}$ |
| $n$ | Exponential factor | | 3.5 |

in the upper mantle. $A_{diff}$, $E_{diff}$ and $V_{diff}$ are the pre-exponential factor, activation energy and activation volume, respectively. $R$ is the gas constant, $T$ the absolute temperature, and $P$ the lithostatic pressure. $\sigma_0$ and $\sigma_{max}$ are surface and maximum yield strength, $f_c$ is the friction coefficient, and $\dot{\varepsilon}_{II}$ the second invariant of the strain rate. A viscosity cut-off is imposed for numerical stability and is $10^{24}$ Pa s, unless mentioned otherwise. The values of all model parameters are listed in Table 1. For more model set-up details, the reader is referred to Agrusta et al.[18]

**Model including non-Newtonian rheology and 60 wt% ol.** The rheological model is assumed to be a combination of diffusion ($\mu_{diff}$), dislocation ($\mu_{disl}$) creep and a pseudo-brittle yield stress rheology ($\mu_y$). The effective viscosity $\mu_{eff}$ is than calculated from the viscosities of the individual mechanisms as:

$$\mu_{eff} = \min(\mu_{diff}, \mu_{disl}, \mu_y) \tag{3.a}$$

with

$$\mu_{disl} = A_{disl}\exp\left(\frac{E_{disl}+PV_{disl}}{nRT}\right)\dot{\varepsilon}_{II}^{\frac{1-n}{n}} \tag{3.b}$$

To facilitate comparison between the models, we choose the rheological parameters to yield a similar average upper and lower mantle viscosity as when only Newtonian rheology is assumed. The rheological parameter values are listed in Table 2. The density contrast ($\Delta\rho$) for the olivine solid–solid phase transitions are reduced to 60% of those in Table 1 to 150 kg m$^{-3}$ for the ol-wd and of 210 kg m$^{-3}$ for the rg-pv+mw.

**Model set-up.** The model domain is 9000 km wide and 3000 km high, and the box is discretized into 2880 × 472 elements, with element sizes ranging from 2.5 to 7.5 km. The grid is refined vertically between 0 and 270 km depth, and horizontally between $x = -5750$ km and $x = 900$ km. The mechanical boundary conditions are free-slip along all boundaries, so only internal buoyancy forces drive the dynamics. The top and bottom thermal boundary conditions are constant temperature, 273 K at the surface and a potential $T_{pot}$ at the bottom. The thermal boundary conditions are different at the left and right boundaries: a zero heat flux is imposed on the left boundary, whereas on the right boundary a mid-ocean ridge (MOR) temperature profile is used to keep the MOR at the model corner (Fig. 6a).

The initial conditions are chosen to represent an overriding and a subducting plate, both with a half-space-cooling thermal structure[62]. The overriding plate extends from a MOR at the upper-right corner to the trench (at $x = 0$ km) with a plate age of 100 Myr. The initial subducting plate has slab with a radius of curvature of 500 km and extends from the trench into the mantle down to a depth of 200 km, which allows self-sustained subduction from the start. The length of the subducting plate, and hence the location of the MOR $x_{MOR}$, are calculated using the initial plate velocity and age such that $x_{MOR} = -V_{SP} \times Age_{SP}$, where $V_{SP}$ is the initial plate velocity and $Age_{SP}$ is the initial subducting plate age at the trench. For each initial subducting plate age and mantle temperature, $V_{SP}$ is determined by solving the instantaneous flow field for $t = 0$. Once $V_{SP}$ is determined, it is used to calculate $x_{MOR}$ for a self-consistent plate age distribution. $V_{SP}$ ranges from ~3 cm per year (lowest $T_{pot}$) to ~25 cm per year (highest $T_{pot}$), and the initial subducting plate age does not significantly influence the initial $V_{SP}$. On top of the entire subducting plate, an 8 km thick low-viscosity layer ($\mu_{weak-layer} = 10^{20}$ Pa s) is present which extends down to 200 km depth to facilitate the decoupling of the converging plates.

**Parameters to simulate early Earth conditions.** Mantle composition, and therefore mantle rheological and phase-transition parameters, are assumed to remain the same from the early Earth to the present day. The mantle potential temperature is varied from its present-day reference $T_{pot} = 1300$ °C by a $\Delta T_{pot}$

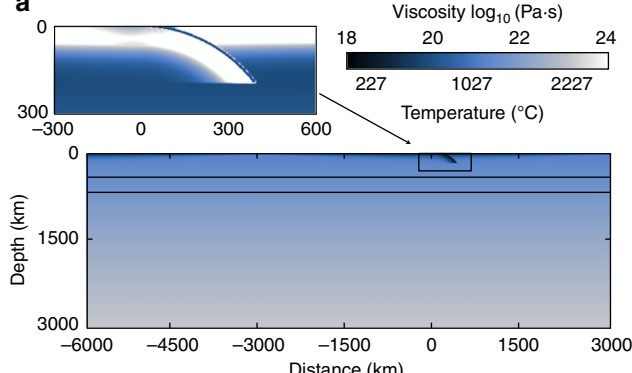

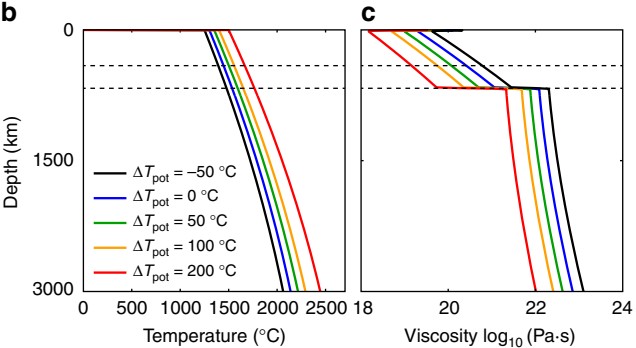

**Fig. 6** Model set-up. **a** Initial condition for a subducting plate 100 Myr old at the reference mantle temperature ($\Delta T_{pot} = 0$ °C). Colours indicate temperature and the horizontal black lines mark the olivine phase transitions. The zoomed area shows corresponding viscosity where the weak layer on top of the plate is visible (dark blue). Background mantle temperature (**b**) and viscosity (**c**) profiles (at the MOR) for the five investigated mantle potential temperatures. The colour scale is used to identify different model cases in the main text, and ranges from black ($\Delta T_{pot} = -50$ °C) for the colder mantle temperature case to red ($\Delta T_{pot} = +200$ °C) for hottest mantle case

between $-50$ °C (further mantle cooling) and $+200$ °C (hotter Earth) (Fig. 6b). Only the olivine solid–solid phase transitions are considered and the density contrasts ($\Delta\rho$) used for the ol-wd (near 410 km depth) and rg-pv + mw (near 670 km) transformations are 250 and 350 kg m$^{-3}$ [Ref. [63]], respectively.

Convective vigour is characterised by the thermal Rayleigh number

$$Ra = \frac{g\alpha_0\rho_0\left(\Delta T + \Delta T_{pot}\right)H^3}{\kappa\mu_0}, \tag{4}$$

where $g$ is the gravitational acceleration, $\alpha_0$, $\rho_0$, $\mu_0$ are the reference thermal expansivity, density and viscosity, respectively, $\kappa$ the thermal diffusivity, $\Delta T + \Delta T_{pot}$ the potential temperature contrast across the box, and $H$ box depth (Table 1). Resistance to sinking through the 670 km phase transition is expressed in terms of the phase buoyancy number,

$$P = \frac{\left(\Delta T + \Delta T_{pot}\right)\gamma_{rw-pr+mw}}{g\rho_0 H}\frac{Rb_{rw-pr+mw}}{Ra}, \tag{5}$$

with phase Rayleigh number

$$Rb_{rw-pr+mw} = \frac{g\Delta\rho_{rw-pr+mw}H^3}{\kappa\mu_0}. \tag{6}$$

And the ol-wd phase transition is implemented similarly.

The rheological parameters (Eq. 2) have been chosen to obtain suggested present-day mantle viscosity values, such that the average upper and lower mantle viscosities are ~$2 \times 10^{20}$ and ~$3 \times 10^{22}$ Pa s, respectively[41,64–68]. At $\Delta T_{pot} = +200$ °C, the viscosity profile reaches average values of ~$1.15 \times 10^{19}$ Pa s and ~$4.6 \times 10^{21}$ Pa s, respectively, for upper and lower mantle (Fig. 6c).

**Data availability**. The outputs of the simulation are available upon request from the corresponding author. The code that supports the findings of this study is available from the corresponding author upon reasonable request.

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

## Acknowledgements
The perceptually uniform colour map oslo is used in this study to prevent visual distortion of the data. Figure 1a has been produced using Generic Mapping Tool, http://gmt.soest.hawaii.edu. We acknowledge funding from the European Research Council (ERC StG 279828) and NERC (grant NE/J008028/1), and the work used the ARCHER UK National Supercomputing Service (http://www.archer.ac.uk) and appreciate the comments of D. Selby, E. Lewellin and J. Baldini on an earlier version of the manuscript. Data for this paper can be made available upon request from the authors.

## Author contributions
R.A., J.v.H. and S.G. developed the concepts of the study and contributed to the writing of the manuscript. R.A. developed and analysed the models.

## Additional information

**Competing interests:** The authors declare no competing interests.

