## [Peer Review File · Nature Communications]

Reviewers' comments:

Reviewer #1 (Remarks to the Author):

This study claims, against the common believe, that mantle convection on Early Earth might have actually been less layered, and that the whole mantle was therefore mixed more efficiently. This is a provocative claim that certainly is of interest to the whole Earth Science community, but clearly also to many other fields beyond, as it potentially resets our current view on many crucial and strongly debated aspects regarding planet Earth's evolution.

This authors' claim is based on a state-of-the-art, well tested numerical code. The presented suite of models covers a wide potential range of unknown subduction and mantle-convection parameters and hence is a solid base to draw above conclusions.

In addition, the numerical and physical model is well described and should therefore be reproducible by other researchers.

I have only two minor, but important, remarks about the presentation and description of these interesting findings. After these adjustments, I fully recommend the manuscript by Agrusta et al. for publication in Nature Communication.

Main comment 1:

As is the case for many other studies too, the terminology in this manuscript falsely suggests that slabs either freely penetrate or fully stagnate at the 670 km phase transition. However, there is clearly a more continuous spectrum ranging from little deflection to strong deflection, as every slab experiences a dynamic impact when crossing the transition zone, and it is unlikely that a slab can remain fully stagnant at the transition zone. I think it is therefore important to make this clear to the reader to prevent wrong conceptions of such an important dynamic aspect of mantle convection.

Main comment 2:

The use of a non-scientific rainbow colour map:

It is unfortunate that studies using one of the most unscientific colour maps are still submitted (and sometimes accepted). The used colour map adds a non-negligible error to the presented data, is fully unreadable or, at best, misleading readers, and clearly should not be used here or elsewhere in a scientific context.

Below I outline some of the specific problems to the current manuscript.

Figure 2: regional viscosity variations, like small scale up- and downwellings and the important effects of the power-law rheology in the upper-mantle, are totally hidden in the greenish parts. Moreover, the artificial boundaries introduced to the underlying data with this colour map (e.g., in the reddish part) very likely misrepresent the actual plate thickness and the viscosity difference between upper- and lower mantle. A perceptually-uniform colour map (available for example here: www.fabiocramer.ch/visualisation) would fix the visual error introduced with this particularly unscientific colour scheme used here.

Figure 3: same applies here. In addition, comparing the different panels is misleading because the same parts of the different models are represented with different, unequal parts of the rainbow colour map; upper-mantle small scale structures in panels (d) are much better visible than they are (or would be) in panels (a) for example.

I stop here with my detailed description: It should be clear by now - to both authors and editor - that the rainbow colour map cannot be used for scientifically accurate representations, especially of parameter-field data like the ones shown here. More information about this specific colour map and possible alternatives can, for example, be found here:

<https://blogs.ehu.es/divisions/gd/2017/08/23/the-rainbow-colour-map/> .

Minor comments:

The manuscript refers repeatedly to layered/non-layered convection but in fact only considers the downwelling part (i.e., only one half) of the convective cycle. As the study does not suggest a different behaviour on the likelihood of upwellings to cross the upper-lower mantle boundary in a hotter planet, the assumed importance/dominance of slabs for the convective cycle should be discussed briefly for clarity.

line 12: are some slabs really “trapped” in the upper mantle; I don’t think so. Reformulating this or saying something like “are temporally trapped” would be less confusing to the reader, I think. (see main comment 1).

line 17-19, and throughout the manuscript: again, I don’t think slabs should be divided into either fully stagnating or fully penetrating. All of them are a mix of both, some are deflected only a little, while others are deflected strongly, and remain, at least for a while, in the upper mantle. Even the authors’ example of a “stagnant” slab suggests that its oldest portions reach into the lower mantle. I suggest therefore a rewording like: “temporal stagnation” and “direct penetration”. (see main comment 1)

line 21: does that mean that a lower-density slab fosters lower mantle penetration in general? In other words, is the effect of slab density at the trench (trench retreat rate) more important than at the transition zone (slab-transition zone buoyancy contrast)? Might need clarification if that is not necessarily the case.

line 39: shouldn’t it say: “intermediate/whole layered one” instead?

line 95: shouldn’t it say: “a factor of 10” and “a factor of 40” instead of just “10” and “40”?

line 130: I could not read/see the special characters printed there.

Figure 3: it is not clear that the two panels side by side are two different time snapshots. I suggest adding an indicator (e.g., a time arrow) to the figure and mention it in the caption.

lines 200-206: Can the authors give a possible explanation for the assumption of slab weakening at these depths? Otherwise, this sudden weakening appears as a random, artificial choice to make your model produce a certain behaviour.

lines 262-264: Can the authors at least clearly state under what (dynamic) conditions this quite speculative statement becomes valid (e.g., the location of the slabs entering into the lower mantle needs to change significantly over time)? The cited models still involve quite some uncertainty and assumptions related to lower mantle convection and compositional aspects and there are other models suggesting ways to stabilise the convective pattern in the lower mantle even in case of slabs penetrating into the lower mantle (e.g., Ballmer et al., 2017, Nat. Geosci.).

Fabio Crameri

Reviewer #2 (Remarks to the Author):

The manuscript approaches the problem of the style of mantle convection throughout Earth's history by examining how the style of subduction might have been different in a past hotter mantle. This is certainly an interesting question and one that has important implications for many related fields including the geodynamo, interpretation of large low velocity provinces in the lower mantle, and the time-dependent nature of plate tectonics.

The conclusions of the paper in regards to how subduction would be different in an earlier, hotter earth are based on the assumption that the models presented accurately represent the fundamental processes controlling the dynamics of subducting slabs sinking through the transition and lower mantle in the present-day earth. However, I would argue that several simplifications or assumptions made in the numerical models are not consistent with what is known about present-day earth structure, namely the rheology and the phase transition structure. In addition, there is still considerable debate in the geodynamics community as to what essential process(es) are required to create the bi-modal behavior of slabs in the present-day earth: this uncertainty is completely ignored in the manuscript.

First, all geodynamics models make simplifications or assumptions, but care must be made in then understanding the limitations of the models in terms of interpreting behavior in terms of what is happening in the earth or extrapolating the behavior under different conditions. The models presented assume the mantle viscosity is Newtonian, and uses a yield stress and viscosity cut-off to limit the viscosity in cold regions. It is well-known that the upper mantle deforms by non-Newtonian mechanism of dislocation creep (e.g., because we observe seismic anisotropy and because geodetic observations are require non-linear viscosity to match time-dependent relaxation) as is predicted by laboratory observations for olivine (e.g., Karato and Wu, *Science*, 1996; Kohlstedt and Hirth, *Sub. Factory*, 2003, ...)

Is it important to include the effect of non-Newtonian viscosity in subduction models – for some studies, maybe not, but for this manuscript I think it is essential because you are also varying the temperature of the mantle, which also affects the viscosity. Non-Newtonian viscosity is most likely to affect the viscosity in the asthenosphere and around subducting slabs because these are the locations of large stresses (or strain-rates). It is exactly the viscosity in these regions that controls how the slabs sinks through upper mantle AND whether the flow induced by this sinking will lead to trench retreat or trench advance (see also Holt and Becker, *GJI* 2017).

If the viscosity, in these key regions in the present day upper mantle, are already very low ($< 10^{19}$ Pa s) because of non-Newtonian viscosity, then increasing the temperature of the mantle in the past will not lead to the same change in trench motion behavior as you have found in your models (this would also be complicated by the effect of grain-size and how this controls the transition between diffusion and dislocation creep). Because in your models the trench motion controls whether slabs sink into the lower mantle or are trapped in the transition zone, the fact that the models are missing this important, known behavior of the mantle is a problem.

Second, in regards to the phase transitions that affect slab. In your simplified models, you assume that the whole mantle is made of olivine and that the only phase transitions are olivine-wadsleyite at 410 km and ringwoodite-bridgmanite+ferropericlasite 660 km. While this was a common simplified model used in the past, several studies have shown that this model over-predicts the effect of the phase transitions because in the real earth, olivine is only 60% of the composition and that both the other phase transition in olivine and the phase in the pyroxene counter-act the effect of the phase transition at 660 km (see for example the compositionally-dependent phase transition model used in Arredondo and Billen, J. of Geodyn 2016 and JGR 2017). In addition, the more recent and more robust laboratory experiments for the clapeyron slope of the ringwoodite-bridgmanite+ferropericlasite transition have shown that only the smallest value of around -1 MPa/K are appropriate, greatly diminishing the effect of this phase transition.

Is this simplification important for the conclusions of this study. Yes, because the whole argument is based on the competition between convective vigor (Rayleigh number) and the resistance to sinking into the lower mantle caused by the ringwoodite-bridgmanite+ferropericlasite phase transition (Phase buoyancy). Because your model over-predicts the effect of phase buoyancy, the conclusions or interpretation based on these models may not be correct.

Related to the issue of the phase transitions model used in the simulations, is the argument made in the paper that in the present-day earth it is this phase transition that is responsible for the observed bimodal behavior of slabs based on their age. While the observation that there is a bimodal behavior of slabs that is related to slab age, the same papers that you cite as a references for this observation present models that show that there are other processes that could be responsible for causing this bimodal behavior (metastable olivine or metastable pyroxene). So rather than it being well-established that the ringwoodite-bridgmanite+ferropericlasite phase transition is responsible for the bimodality, there is in fact quite a bit of debate as to what causes this behavior. Omitting this information from the paper ignores the current state of understanding and debate in the discipline.

Finally, I'll make an additional note about the reference "present-day" viscosity structure in the model. In the paper, you state the parameters are chosen such that the radial viscosity model meets the constraints that the average upper mantle viscosity is 2×10^{20} Pa s and average lower mantle viscosity of $\sim 3 \times 10^{22}$ Pa s. While these values are consistent geoid constraints and what people generally use in models, there are much better data and references to point to than the references you provide. For example, from Mitrovica, JGR 1996:

We derive a constraint on the "average" viscosity of the mantle of $0.65-1.10 \times 10^{21}$ Pa s, where the "average" resolved by the data encompasses a region which extends from the base of the lithosphere to a depth of 1400 km. This indicates that many previous analyses which have invoked the Haskell value

of 10^{21} Pa s as a constrain on the average upper mantle (i.e., above 670 km depth) viscosity alone have misinterpreted the resolving power of the inference.

This is a very robust constraint, which strongly limits how high the upper-most lower-mantle viscosity can get and requires that upper mantle, below the lithosphere, balance any increase in viscosity in the lower mantle. Your present-day profile appears to meet this constraint, but the reference you site, don't actually provide this kind of strong constraint.

Another very useful summary reference is Burgmann & Dresen, AREPS 2008: they summarize a variety of geodetic constraints showing that the shallow part of the upper mantle (<160 km or so) has a viscosity of $<10^{18}$ to 10^{19} Pa s in several different locations including in present-day subduction zones (Alaska, Cascadia). Finally, both they (and the paper that you site by Karato and Wu, Science 1996) point out that there is clear evidence in many different tectonic environments that the upper mantle (at least above the 410-km) deforms with non-Newtonian viscosity.

It is these constraints on the shallow mantle in actively deforming regions that strongly support the importance of including non-Newtonian viscosity, and allowing the viscosity in these actively deforming areas to be significantly lower (10^{18} to 10^{19} Pa s) than the value found in the deeper mantle in cratonic regions ($5 - 10 \times 10^{20}$ Pa s). Your own models show that if you allow for this kind of non-linear weakening around slabs and beneath the asthenosphere, then your present-day models would behave like your hotter-earth models.

- Magali Billen

Reviewers' comments:

Reviewer #1 (Remarks to the Author):

This study claims, against the common believe, that mantle convection on Early Earth might have actually been less layered, and that the whole mantle was therefore mixed more efficiently. This is a provocative claim that certainly is of interest to the whole Earth Science community, but clearly also to many other fields beyond, as it potentially resets our current view on many crucial and strongly debated aspects regarding planet Earth's evolution.

This authors' claim is based on a state-of-the-art, well tested numerical code. The presented suite of models covers a wide potential range of unknown subduction and mantle-convection parameters and hence is a solid base to draw above conclusions.

In addition, the numerical and physical model is well described and should therefore be reproducible by other researchers.

I have only two minor, but important, remarks about the presentation and description of these interesting findings. After these adjustments, I fully recommend the manuscript by Agrusta et al. for publication in Nature Communication.

We thank Fabio Cramer for his thoughtful and constructive comments.

Main comment 1:

As is the case for many other studies too, the terminology in this manuscript falsely suggests that slabs either freely penetrate or fully stagnate at the 670 km phase transition. However, there is clearly a more continuous spectrum ranging from little deflection to strong deflection, as every slab experiences a dynamic impact when crossing the transition zone, and it is unlikely that a slab can remain fully stagnant at the transition zone. I think it is therefore important to make this clear to the reader to prevent wrong conceptions of such an important dynamic aspect of mantle convection.

We agree with the reviewer that the range of slab-transition zone interactions also includes slabs that are neither strictly penetrating nor fully stagnant, and our measure D_{TZ}/D_{LM} captures this continuum. We used, as the reviewer notes, the commonly used terminology of penetrating or stagnating to classify the slabs according to the end-member morphologies we observe on Earth today. In our review paper [Goes et al., 2017], we present a detailed description of slab transition zone interaction, including discussion of the time scales of stagnation.

We refer to this paper in the text, and in addition, based on the reviewer suggestions, we now describe the endmember scenarios more cautiously (using terms like flattening, temporarily stagnant and by adding time scales of stagnation) to not leave the impression that stagnation is permanent. Some of the main changes are in the first line of the abstract (flattening rather than stagnant), in line 82 in the introduction (addition of a time scale of stagnation) and in the wording throughout the section on present-day dynamics (line 125-148).

Main comment 2:

The use of a non-scientific rainbow colour map:

It is unfortunate that studies using one of the most unscientific colour maps are still submitted (and sometimes accepted). The used colour map adds a non-negligible error to the presented data, is fully unreadable or, at best, misleading readers, and clearly should not be used here or elsewhere in a scientific context.

Below I outline some of the specific problems to the current manuscript.

Figure 2: regional viscosity variations, like small scale up- and downwellings and the important effects of the power-law rheology in the upper-mantle, are totally hidden in the greenish parts.

Moreover, the artificial boundaries introduced to the underlying data with this colour map (e.g., in the reddish part) very likely misrepresent the actual plate thickness and the viscosity difference between upper- and lower mantle. A perceptually-uniform colour map (available for example here: www.fabiocrameri.ch/visualisation) would fix the visual error introduced with this particularly unscientific colour scheme used here.

Figure 3: same applies here. In addition, comparing the different panels is misleading because the same parts of the different models are represented with different, unequal parts of the rainbow colour map; upper-mantle small scale structures in panels (d) are much better visible than they are (or would be) in panels (a) for example.

I stop here with my detailed description: It should be clear by now - to both authors and editor - that the rainbow colour map cannot be used for scientifically accurate representations, especially of parameter-field data like the ones shown here. More information about this specific colour map and possible alternatives can, for example, be found here: <https://blogs.egu.eu/divisions/gd/2017/08/23/the-rainbow-colour-map/>.

We thank the reviewer for this suggestion and we remade the figures representing the viscosity field (Fig. 2, 3, 5 and Supplementary Fig. S2) using one of his suggested colourscales, a black through blue to white colormap.

Minor comments:

The manuscript refers repeatedly to layered/non-layered convection but in fact only considers the downwelling part (i.e., only one half) of the convective cycle. As the study does not suggest a different behaviour on the likeliness of upwellings to cross the upper-lower mantle boundary in a hotter planet, the assumed importance/dominance of slabs for the convective cycle should be discussed briefly for clarity.

In the manuscript we already mentioned that upwellings likely contribute only 10-20% to the heat transport (Introduction). However, as the reviewer points out, we did not discuss the role of upwellings at hotter conditions, where they may have a larger contribution to the transport of heat. Upwellings would however readily cross the phase transition at hotter mantle conditions, because for transition-zone temperatures higher than 2000 °C for a pyrolite or harzburgite composition the transition to perovskite at the base of the upper mantle becomes exothermic (positive Clapeyron slope) [e.g., review by *Faccenda and Dal Zilio, 2017*] which facilitates material flow through the phase transition.

We now mention this point in the discussion of the revised version (line 301-308). The new text reads: *"The presented study does ignore the active upwelling part of the global convection. As mentioned in the introduction, upwellings probably contribute only ~10-20 % of present-day mantle flux^{3,4}. Furthermore, upwellings are expected to readily cross the phase transition at hotter mantle conditions because for transition-zone temperatures higher*

than 2000°C, the transition in a pyrolite or harzburgite composition to postspinel phases at the base of the upper mantle becomes exothermic (positive Clapeyron slope)⁵³ which facilitates material flow through the phase transition. “

line 12: are some slabs really “trapped” in the upper mantle; I don’t think so. Reformulating this or saying something like “are temporally trapped” would be less confusing to the reader, I think. (see main comment 1).

In the revised manuscript we modified this term in the abstract to “flattened”.

line 17-19, and throughout the manuscript: again, I don’t think slabs should be divided into either fully stagnating or fully penetrating. All of them are a mix of both, some are deflected only a little, while others are deflected strongly, and remain, at least for a while, in the upper mantle. Even the authors’ example of a “stagnant” slab suggests that its oldest portions reach into the lower mantle. I suggest therefore a rewording like: “temporal stagnation” and “direct penetration”. (see main comment 1)

We revised wording throughout the manuscript to avoid this impression (see our response to comment 1), although we note that the term stagnant by itself does not mean that something is immobile forever.

line 21: does that mean that a lower-density slab fosters lower mantle penetration in general? In other words, is the effect of slab density at the trench (trench retreat rate) more important than at the transition zone (slab-transition zone buoyancy contrast)? Might need clarification if that is not necessarily the case.

Indeed, lower density in general should facilitate slab penetration due to both reducing trench mobility and reducing the phase buoyancy contrast. However, in the context of the sentence we would like to highlight the aspect of the trench mobility that we show in this paper is key for the whole dynamics.

We clarify this point further in the introduction (lines 80-83): “In these models^{2,19–22}, stronger and denser (old) slabs interacting with both an endothermic phase change and viscosity increase induce trench retreat and stagnate (at least for 10s to 100s of m.y.), while weaker and lighter (young) slabs accumulate at relatively stationary trenches, which aids penetration.”

line 39: shouldn’t it say: “intermediate/whole layered one” instead?

Yes thanks, we corrected the sentence to say “a system intermediate between whole and layered”.

line 95: shouldn’t it say: “a factor of 10” and “a factor of 40” instead of just “10” and “40”?

We corrected this in the revised version

line 130: I could not read/see the special characters printed there.

It is “and”. We checked that in the revised version it is readable.

Figure 3: it is not clear that the two panels side by side are two different time snapshots. I

suggest adding an indicator (e.g., a time arrow) to the figure and mention it in the caption.

In the revised manuscript, we added the time arrow and indicate the time of each of the snapshots and mention that there are two snapshots in the caption, like the reviewer suggested.

lines 200-206: Can the authors give a possible explanation for the assumption of slab weakening at these depths? Otherwise, this sudden weakening appears as a random, artificial choice to make your model produce a certain behaviour.

Such weakening has been proposed to be the result of grain-size reduction that occurs when the slab goes through the olivine-wadsleyite transition, which could make the core of the slab deform in diffusion creep [Karato *et al.*, 2001]. We find that the behaviour of such weak-transition zone slabs is similar to that previously found in models with isoviscous slabs. Hence, we infer from these models that at higher temperatures when the slabs weaken further, they would start to behave like those in previous models where hotter slabs become more prone to stagnation, and younger, weaker slabs start to stagnate more readily than older, stronger slabs.

We now mention this weakening mechanism in the revised manuscript (line 220-222): *“Slab weakening in the transition zone, which can be due to grain size reduction during phase transformation, has been previously proposed to lead to slab stagnation¹⁷.”*

lines 262-264: Can the authors at least clearly state under what (dynamic) conditions this quite speculative statement becomes valid (e.g., the location of the slabs entering into the lower mantle needs to change significantly over time)? The cited models still involve quite some uncertainty and assumptions related to lower mantle convection and compositional aspects and there are other models suggesting ways to stabilise the convective pattern in the lower mantle even in case of slabs penetrating into the lower mantle (e.g., Ballmer *et al.*, 2017, *Nat. Geosci.*).

We assume the vigour of mantle convection was higher in the past and our straightforward conclusion is that high Ra convection enhances mantle mixing. The papers we cite have investigated such mixing in various thermo-chemical systems, and show that under most of the parameter ranges they investigated (e.g., various chemical buoyancy ratios, Clapeyron slopes), enhanced vigour of convection makes it difficult to sustain piles over the age of the Earth. Ballmer *et al.* (2017) suggests that assuming primordial mantle with a (quite speculative) composition with a higher fraction of the bridgmanite than the usually assumed pyrolitic mantle composition, a substantial fraction of the lower mantle can remain unmixed to the present day, if there is a sufficient chemical viscosity contrast. We already reference the paper by Deschamps and Tackley (2009), who showed something similar, namely that although it is difficult to maintain piles under many conditions, if they also have a distinct and sufficiently high chemically controlled viscosity contrast, this can stabilise them. The paper by Ballmer shows that maintaining heterogeneity in the core of the lower mantle convection cells can also be done with sufficiently high viscosity contrast. However, our point here pertains to piles, so we feel the Deschamps and Tackley reference is quite an appropriate one, and there is not really the space to expand the discussion to other distributions of chemical heterogeneity.

Fabio Crameri

Reviewer #2 (Remarks to the Author):

The manuscript approaches the problem of the style of mantle convection throughout Earth's history by examining how the style of subduction might have been different in a past hotter mantle. This is certainly an interesting question and one that has important implications for many related fields including the geodynamo, interpretation of large low velocity provinces in the lower mantle, and the time-dependent nature of plate tectonics.

We thank Magali Billen for her thorough and thoughtful review that helped us clarify several key aspects of the paper.

The conclusions of the paper in regards to how subduction would be different in an earlier, hotter earth are based on the assumption that the models presented accurately represent the fundamental processes controlling the dynamics of subducting slabs sinking through the transition and lower mantle in the present-day earth. However, I would argue that several simplifications or assumptions made in the numerical models are not consistent with what is known about present-day earth structure, namely the rheology and the phase transition structure.

We agree with the reviewer that the chosen rheology in our models is relatively simple, and that further discussion of the used rheology might be insightful. In previous work, we contributed to the debate about the role of the large number of rheological parameters on slab dynamics, e.g. through the importance of metastable phases [Agrusta *et al.*, 2014] and slab rigidity [Cizkova *et al.* 2002, Agrusta *et al.*, 2017]. To reduce this wide parameter range, we decided to focus on the available unambiguous evidence that, compared to young slabs, old plates tend to sink under faster trench retreat, allowing slab stagnation in the transition zone [Goes *et al.*, 2008, 2017]. However, we concur with the reviewer that adding some of the debate in the manuscript is useful to provide the reader more background. Also, we used a linear rheology and a pure olivine composition for the mantle. These simplifications are worth testing to increase the confidence of the models. We addressed both these concerns in the response to the related comments made by the reviewer below.

(1) In addition, there is still considerable debate in the geodynamics community as to what essential process(es) are required to create the bi-modal behavior of slabs in the present-day earth: this uncertainty is completely ignored in the manuscript.

As the reviewer highlights there has been substantial debate about the processes controlling the bi-modal slab transition zone dynamics. What has become clear from the many studies that have been done (see e.g. our recent review [Goes *et al.*, 2017]) is that slab stagnation requires an interplay of factors, including (i) a viscosity increase which leads to an increased resistance to sinking causing slab deformation within the transition zone, (ii) an endothermic phase transition which adds an additional resistance that can lead to significant accumulation of material, rather than just a slowed sinking, and finally (iii) trench migration which allows for slab flattening rather than just thickening or buckling.

For the bi-modal slab dynamics observed today, there is a very strong correlation between slabs that are flattened in the transition zone and a history of trench migration [e.g. Van der Hilst and Seno, 1993; Van der Hilst *et al.* 1995; Goes *et al.*, 2017]. Furthermore, most flattened slabs occur where older plates have subducted [e.g., Karato *et al.*, 2001, King *et al.*, 2015]. So, the mechanism responsible for bi-modal subduction behavior is linked to that governing variable trench migration and it needs to explain why older plates have a

stronger propensity to migrate and stagnate than younger plates. Many models have shown that the higher densities and strengths expected due to lower temperatures of older plates enhance trench retreat, so these basic plate properties can explain the observations of bi-modal behavior in a slab interacting with a viscosity and phase boundary.

In addition, it has been proposed that the presence of metastable phases in the transition zone, which are expected to persist preferentially in older, colder slabs will increase slab buoyancy in the transition zone and further facilitate slab stagnation [Tetzlaff and Schmeling, 2000; Agrusta et al., 2014; King et al., 2015]. Another suggested mechanism is the reduction of slab rigidity in the transition zone. Grain-size reduction during phase transformations can weaken the cold core of slabs and would be most efficient where phase transformations are delayed due to metastability, thus making older slabs weakest in the transition zone. Slab weakening in the transition zone can aid stagnation as well [Karato et al., 2001, Cizkova et al., 2002, Nakuki et al., 2010]. The effects of both metastability and grain-size reduction will be reduced at higher temperature thus these factors would also enhance slab penetration in the past Earth conditions. However, in addition to these mechanisms hampering the sinking of older slabs, the effects of slab age on slab density and strength at the trench are still required to explain the correlation between slab-transition zone dynamics and trench retreat.

So although other mechanisms may also play a role, we start from a model set-up where slab behavior is governed by age-dependent slab strength and density that appear to be the first order mechanism that can explain both the correlation with trench migration and old age of present-day stagnant slabs.

We clarify this further in the introduction where the new paragraph (line 76-89) now reads: “*On Earth, its observed that older (denser and stronger) plates have a higher tendency to produce trench retreat and flat slabs above the upper-lower mantle boundary around ~670 km depth than young plates^{17,18}. This behaviour is reproduced in recent dynamical models where plate boundaries move in response to the slab dynamics. In these models^{2,19-22}, stronger and denser (old) slabs interacting with both an endothermic phase change and viscosity increase induce trench retreat and stagnate (at least for 10s to 100s of m.y.), while weaker and lighter (young) slabs accumulate at relatively stationary trenches, which aids penetration. While other factors, like for example the persistence of metastable phases in the slab’s coldest core and associated slab weakening^{17,23-25}, may additionally hamper the sinking of older slabs through the transition zone, variable plate age at the trench can explain the primary observations of today’s mixed slab-transition-zone dynamics and its relation to trench motion^{2,21}. In this study, we use these calibrated models to re-examine how such more plate-like and mobile slabs behave under hotter mantle conditions.*”

And in the discussion, we add a paragraph about the effects of metastable phases and consequent slab weakening (lines 273-277): “*(iii) Metastable phases inside the coldest slabs have been proposed to contribute to the stagnation of older plates in the transition zone^{17,18,23,25}. However, at higher temperatures both the effects of metastability and concurrent slab weakening due to grain size reduction will be reduced, thus also facilitating slab penetration in a hotter Earth.*”

(2) First, all geodynamics models make simplifications or assumptions, but care must be made in then understanding the limitations of the models in terms of interpreting behavior in terms of what is happening in the earth or extrapolating the behavior under different conditions. The models presented assume the mantle viscosity is Newtonian, and uses a yield stress and viscosity cut-off to limit the viscosity in cold regions. It is well-known that the upper mantle deforms by non-Newtonian mechanism of dislocation creep (e.g., because we observe seismic anisotropy and because geodetic observations are require non-linear viscosity to match time dependent relaxation) as is predicted by laboratory observations for olivine

(e.g., Karato and Wu, Science, 1996; Kohlstedt and Hirth, Sub. Factory, 2003, ...). Is it important to include the effect of non-Newtonian viscosity in subduction models – for some studies, maybe not, but for this manuscript I think it is essential because you are also varying the temperature of the mantle, which also affects the viscosity. Non-Newtonian viscosity is most likely to affect the viscosity in the asthenosphere and around subducting slabs because these are the locations of large stresses (or strain-rates). It is exactly the viscosity in these regions that controls how the slabs sink through upper mantle AND whether the flow induced by this sinking will lead to trench retreat or trench advance (see also Holt and Becker, GJI 2017).

We agree with the reviewer that testing the model behavior with a non-Newtonian viscosity is useful to test the robustness of our hypothesis, and we thank her for this suggestion. Therefore, we performed simulations that include non-Newtonian rheology in the upper mantle for the present day condition and for a hotter mantle. These simulations display dislocation creep in the asthenosphere and around the slab, in agreement with observations and what the reviewer pointed out. The non-Newtonian models show the same behaviour as observed for the Newtonian case, with both young and old slabs penetrating for low phase-transition strength, and bi-modal dynamics at higher Clapeyron slopes. Similar bi-modal dynamics were previously found for composite dislocation-diffusion-Peierls creep models of young and old slabs interacting with a viscosity boundary [Garel *et al.*, 2014], where older slabs also drove more trench retreat resulting in slab flattening in the transition than younger slabs. When starting from a case with bimodal slab-transition zone interaction and running the same cases for a hotter mantle ($\Delta t = 50$ °C), both old and young slabs in our models penetrate into the lower mantle. This confirms that it is a robust feature for both Newtonian and non-Newtonian viscous mantle rheologies that a hotter mantle enhances slab penetration.

In the supplementary information, we added two figures to illustrate these models (Fig. S2 and S3) and a description of modifications of the model set up and the rheology parameters used (Table S2). We refer to the results in the main text. In lines 108-111, we write “*The models presented use a Newtonian rheology and assume a composition of 100 wt% of olivine, but additional models, with a composite non-Newtonian creep and only 60 wt% of olivine, that display the same styles of behaviour are in the Supplement.*”

(3) If the viscosity, in these key regions in the present day upper mantle, are already very low ($< 10^{19}$ Pa s) because of non-Newtonian viscosity, then increasing the temperature of the mantle in the past will not lead to the same change in trench motion behavior as you have found in your models (this would also be complicated by the effect of grain-size and how this controls the transition between diffusion and dislocation creep). Because in your models the trench motion controls whether slabs sink into the lower mantle or are trapped in the transition zone, the fact that the models are missing this important, known behavior of the mantle is a problem.

We agree with the reviewer that very low asthenospheric viscosity will lead to less mobility of the trench, however this will happen whether accounting for the deformation by dislocation creep or not. We also agree that dislocation creep is likely the dominant deformation mechanism in the shallower mantle. Laboratory experiments give a wide range of the rheological parameters, and radial mantle viscosity profiles derived by geophysical observation do not uniquely constrain the viscosities of the uppermost mantle layer, although they agree on an average mantle viscosity [King, 2016]. However, there is also the observational constraint that viscosity cannot be so low that it precludes all trench motion.

Furthermore, we think that the indications that trench mobility regulates the slab dynamics of today's mantle are very compelling (see our response to point (1) and [Agrusta *et al.*, 2017]). Hence, while choosing a set of rheological parameters (in diffusion and dislocation creep) that yield viscosities compatible with the observational constraints [e.g. King, 2016], we also build a present-day model that yields the observed variability in slab-transition zone interaction as a function of plate age.

We now mention the importance of asthenospheric viscosity in lines 190-195: “*Note that these boundaries can shift within the uncertainties and trade-offs between model parameters. At a higher viscosity jump at the base of the transition zone, the field of stagnant and mixed modes expands to lower phase buoyancy and higher Ra. A reduction of the asthenospheric mantle viscosity, leading to less trench mobility^{30,31}, would induce an opposite shift. The main features of the regime diagram as a function of temperature are however robust.*” See also our response to point (2) and point (7).

(4) Second, in regards to the phase transitions that affect slab. In your simplified models, you assume that the whole mantle is made of olivine and that the only phase transitions are olivine-wadsleyite at 410 km and ringwooditebridgmanite+ferropericalse 660 km. While this was a common simplified model used in the past, several studies have show that this model over-predicts the affect of the phase transitions because in the real earth, olivine is only 60% of the composition and that both the other phase transition in olivine and the phase in the pyroxene counter-act the effect of the phase transition at 660 km (see for example the compositionally-dependent phase transition model used in Arredondo and Billen, J. of Geodyn 2016 and JGR 2017). In addition, the more recent and more robust laboratory experiments for the clapeyron slope of the ringwooditebridgmanite+ferropericalse transition have shown that only the smallest value of around -1 MPa/K are appropriate, greatly diminishing the effect of this phase transition.

Is this simplification important for the conclusions of this study. Yes, because the whole argument is based on the competition between convective vigor (Rayleigh number) and the resistance to sinking into the lower mantle caused by the ringwoodite-bridgmanite+ferropericalse phase transition (Phase buoyancy). Because your model over-predicts the effect of phase buoyancy, the conclusions or interpretation based on these models may not be correct.

We agree with this point as well, and we in the new models that include non-Newtonian rheology we also take into account 60 wt% of olivine. These models illustrate that even with such parameters, bimodal slab-transition zone interaction can be reproduced. In Goes *et al.* [2017] we also discussed this point, and estimated that slab stagnation can be reached with 60 wt% of olivine and a Clapeyron slope -1 MPa/K assuming a lower-mantle viscosity slightly higher than used in the models presented here, but still well within the range suggested by observations [King, 2016]. The exact value of the postspinel phase transition Clapeyron slope is still debated and values we assumed in our models fall within the range of recent estimates [e.g., review by Faccenda and Dal Zilio, 2017] but also the reviewer's work [Arredondo and Billen, 2016, 2017].

We added some discussion about the effects of these uncertainties and trade-offs where we present the regime diagram in Fig 4, in the section on the dynamics in a hotter Earth (lines 190-195). And we note that given that we start from a mixed-mode regime in the present-day, increasing mantle temperatures pushes slab behaviour towards penetration for slabs of all ages (line 202-205).

(5) Related to the issue of the phase transitions model used in the simulations, is the argument made in the paper that in the present-day earth it is this phase transition that is responsible for the observed bimodal behavior of slabs based on their age.

While the observation that there is a bimodal behavior of slabs that is related to slab age, the same papers that you cite as a reference for this observation present models that show that there are other processes that could be responsible for causing this bimodal behavior (metastable olivine or metastable pyroxene). So rather than it being well-established that the ringwoodite-bridgmanite+ferropericlasite phase transition is responsible for the bimodality, there is in fact quite a bit of debate as to what causes this behavior. Omitting this information from the paper ignores the current state of understanding and debate in the discipline.

We agree with the reviewer, and now discuss the possible range of causes for the observed slab bi-modality. See our response to the first comment for changes made.

(6) Finally, I'll make an additional note about the reference "present-day" viscosity structure in the model. In the paper, you state the parameters are chosen such that the radial viscosity model meets the constraints that the average upper mantle viscosity is 2×10^{20} Pa s and average lower mantle viscosity of $\sim 3 \times 10^{22}$ Pa s. While these values are consistent geoid constraints and what people generally use in models, there are much better data and references to point to than the references you provide. For example, from Mitrovica, JGR 1996: We derive a constraint on the "average" viscosity of the mantle of $0.65-1.10 \times 10^{21}$ Pa s, where the "average" resolved by the data encompasses a region which extends from the base of the lithosphere to a depth of 1400 km. This indicates that many previous analyses which have invoked the Haskell value of 10^{21} Pa s as a constraint on the average upper mantle (i.e., above 670 km depth) viscosity alone have misinterpreted the resolving power of the inference. This is a very robust constraint, which strongly limits how high the upper-most lower-mantle viscosity can get and requires that upper mantle, below the lithosphere, balance any increase in viscosity in the lower mantle. Your present-day profile appears to meet this constraint, but the reference you cite, don't actually provide this kind of strong constraint.

We thank the reviewer for this suggestion, and now include the reference to Mitrovica [1996].

(7) Another very useful summary reference is Burgmann & Dresen, AREPS 2008: they summarize a variety of geodetic constraints showing that the shallow part of the upper mantle (<160 km or so) has a viscosity of $<10^{18}$ to 10^{19} Pa s in several different locations including in present-day subduction zones (Alaska, Cascadia).

Finally, both they (and the paper that you cite by Karato and Wu, Science 1996) point out that there is clear evidence in many different tectonic environments that the upper mantle (at least above the 410-km) deforms with non-Newtonian viscosity.

It is these constraints on the shallow mantle in actively deforming regions that strongly support the importance of including non-Newtonian viscosity, and allowing the viscosity in these actively deforming areas to be significantly lower (10^{18} to 10^{19} Pa s) than the value found in the deeper mantle in cratonic regions ($5-10 \times 10^{20}$ Pa s). Your own models show that if you allow for this kind of non-linear weakening around slabs and beneath the asthenosphere, then your present-day models would behave like your hotter-earth models.

We agree with the reviewer that the shallow mantle probably deforms with non-Newtonian rheology. But we argue that the very low viscosities mentioned by the reviewer may not be representative of the asthenosphere in general. First, James *et al.* [2000; 2009] and

Bürmann and Dresen [2008] show that the glacial isostatic adjustment data for the Cascadia subduction zone can fit a wide range of asthenospheric thicknesses (from 140 km to 380 km) and viscosities from 3×10^{18} Pa s to 4×10^{19} Pa s. In addition, some of the locations discussed in [*Bürmann and Dresen*, 2008] might not be representative for typical upper mantle viscosities. E.g., Iceland might have reduced viscosities due to elevated mantle temperatures. Indeed, our models show that if viscosities as low as 10^{18} Pa s are representative for the entire shallow upper mantle, then slabs in our models don't show any significant rollback, which doesn't support the observations that trench migration is a ubiquitous feature of plate tectonics. However, with viscosities on the order of 10^{19} Pa s sufficient trench retreat is possible to allow for bi-modal slab dynamics. In addition, with some other changes in model parameters (e.g., strength and thickness of the layer decoupling the two plates), the asthenospheric viscosity at which the behaviour changes can shift.

Regardless of these uncertainties and trade-offs between model parameters, we want to start the analysis from a model that reproduces today's bi-modal behaviour with older slabs stagnating and younger slabs penetrating. The basic mechanics of such models are understood and reproduced by a number of different model set-ups (both analogue and numerical) [see discussion in *Goes et al.* 2017] In this paper we show that under hotter mantle conditions, such models predict a shift to a mode where all slabs will penetrate.

- Magali Billen

REVIEWERS' COMMENTS:

Reviewer #1 (Remarks to the Author):

The authors have carefully addressed and/or implemented all my comments and I have no other points to raise, and therefore recommend publication of the revised manuscript as is.

Fabio Crameri

Reviewer #2 (Remarks to the Author):

The authors have done a very good job responding to both reviewers comments and have done a good job addressing these comments in the paper in the limited space available in a Nature Comm. paper. The models added to the supplemental information address the main concern I had on the manuscript. I have not further comments or suggestions and I recommend that the manuscript be published.

REVIEWERS' COMMENTS:

Reviewer #1 (Remarks to the Author):

The authors have carefully addressed and/or implemented all my comments and I have no other points to raise, and therefore recommend publication of the revised manuscript as is.

Fabio Cramerì

Reviewer #2 (Remarks to the Author):

The authors have done a very good job responding to both reviewers comments and have done a good job addressing these comments in the paper in the limited space available in a Nature Comm. paper. The models added to the supplemental information address the main concern I had on the manuscript. I have not further comments or suggestions and I recommend that the manuscript be published.

We thank the reviewers for their thoughtful and constructive review that helped us clarify several key aspects of the paper.